# Consumer Views and Experiences of Secondary-Care Services Following REFOCUS-PULSAR Staff Recovery-Oriented Practices Training

**DOI:** 10.3390/ijerph20105894

**Published:** 2023-05-20

**Authors:** Michelle Kehoe, Ellie Fossey, Vrinda Edan, Lisa Chaffey, Lisa Brophy, Penelope June Weller, Frances Shawyer, Graham Meadows

**Affiliations:** 1Department of Occupational Therapy, School of Primary and Allied Health Care, Monash University, Peninsula Campus Building G, 47–49 Moorooduc Hwy, Frankston, VIC 3199, Australia; 2Alfred Health, Adult Mental and Addiction Health, 55 Commercial Road, Melbourne, VIC 3004, Australia; 3Centre for Mental Health Nursing, Department of Nursing, School of Health Sciences, The University of Melbourne, Melbourne, VIC 3010, Australia; 4Social Work and Social Policy, Department of Community and Clinical Health School of Allied Health, Human Services and Sport, La Trobe University, Melbourne, VIC 3086, Australia; 5The Centre for Mental Health, Melbourne School of Population and Global Health, The University of Melbourne, Melbourne, VIC 3000, Australia; 6College of Business and Law, RMIT University, Melbourne, VIC 3000, Australia; 7Southern Synergy, Monash Health, Dandenong, VIC 3175, Australia; 8Department of Psychiatry, School of Clinical Sciences at Monash Health, Monash University, Melbourne, VIC 3168, Australia; 9Monash Centre for Health Research and Implementation, Monash University, Clayton, VIC 3168, Australia; 10School of Primary and Allied Health Care, Monash University, Melbourne, VIC 3168, Australia

**Keywords:** mental health recovery, recovery-oriented practices, community mental health care, consumers, personal recovery

## Abstract

Background: The use of recovery-oriented practice (ROP) can be challenging to implement in mental health services. This qualitative sub-study of the Principles Unite Local Services Assisting Recovery (PULSAR) project explored how consumers perceive their recovery following community mental health staff undertaking specific ROP training. Methods: Using a qualitative participatory methodology, 21 consumers (aged 18–63 years) participated in one-on-one interviews. A thematic analysis was applied. Results: Four main themes were extracted: (1) connection, (2) supportive relationships, (3) a better life, and (4) barriers. Connections to community and professional staff were important to support consumers in their recovery journey. Many consumers were seeking and striving towards a better life that was personal and individual to each of them, and how they made meaning around the idea of a better life. Barriers to recovery primarily focused on a lack of choice. A minor theme of ‘uncertainty’ suggested that consumers struggled to identify what their recovered future might entail. Conclusion: Despite staff undertaking the ROP training, all participants struggled to identify language and aspects of recovery in their interaction with the service, suggesting a need for staff to promote open, collaborative conversations around recovery. A specifically targeted recovery resource might facilitate such conversation.

## 1. Introduction

Recovery and recovery-oriented practice have become a focus for mental health policy and practice internationally [1]; there are also concerns that the term ‘recovery’ is misunderstood or co-opted by services and, so, the implementation of genuine recovery-oriented practices may lag behind the vision [2]. Originally conceived by, and for, people with experiences of mental health issues and service use, recovery is a lived experience [2,3]; its meaning, therefore, is diverse, personal, and multi-dimensional [4]. While there can be no universal definition, several recovery processes have been described, including connectedness, hope, identity, meaning, and empowerment, in efforts to facilitate the translation of recovery-oriented mental health policy into practices [5]. Recent reviews of qualitative accounts and narratives further highlight that recovery is a highly individual meaning-making process, and may include faith or spirituality, social, political, and human rights dimensions that generate or hinder growth for an individual [4,6].

While recovery is a term used to describe a lived experience, recovery-oriented practice (ROP) refers to principles for how services provide mental health care. ROP identifies practices through which practitioners provide supports and interventions to facilitate individuals in recovery. Broadly, ROP is a collaborative approach through which the lived experience, choices, and rights of consumers are valued and respected to support consumers to live personally meaningful lives [7], noting that the term ‘consumer’ is used here to refer to a person with lived or living experience of a diagnosis of mental health issues. ROP, then, is in contrast to a clinical focus often determined by professionals [1]. Several approaches to implementing recovery-oriented practice have been developed; they generally emphasize respectful, collaborative relationships, and practices focused on values, strengths, fostering hope and self-determination, finding meaning and purpose, and developing supports to facilitate recovery, thriving, and wellbeing [5,7,8,9].

Little research has investigated consumer views and experiences of recovery-oriented mental health care [10]. Consumers report highly valuing caring and trusting relationships with mental health staff, and these relationships may act either as a facilitator or barrier to recovery [6,10]. Relationships considered helpful in recovery are those where consumers spend time with professionals who they know and trust, who support them especially around decision-making, collaborating with them, have meaningful engagement, and keep them informed [7,10,11,12]. Furthermore, when consumers feel supported in their recovery, they enjoy greater levels of wellbeing [13]. Fewer studies have sought consumer views in evaluating the success of mental health staff receiving specific training in recovery-oriented practices, with mixed results. For instance, Wilrycx et al. [14] reported mental health service users did not rate their relationships with staff as more recovery-oriented following staff training based on the REFOCUS approach [15], whereas a qualitative study by Wallace and colleagues [16] found that when the REFOCUS pro-recovery practices were successfully implemented, consumers gained knowledge of their strengths and a more hopeful perspective. To further explore consumer views of their mental health care following mental health staff training in recovery-oriented practices based on the REFOCUS framework, the current qualitative study was designed as part of the REFOCUS–PULSAR specialist mental health care trial conducted in Australia [17]. In this stepped-wedge cluster randomized controlled trial (RCT) of ROP training, the REFOCUS training materials [15] were adapted to suit the Australian mental health service context, following which staff across 14 clinical and non-clinical specialist community mental health services in Melbourne’s south-eastern region received training. In total, 190 staff were trained in ROP by two trainers, one of whom was a person with lived experience of a diagnosis of mental illness [17]. This trial found a small, but significant effect, with consumers showing greater improvements in their recovery, as measured by the Questionnaire about the Process of Recovery (QPR) [18], where staff had undertaken the REFOCUS–PULSAR training compared to consumers from services where staff had not undertaken this training [17]. Hence, it provides evidence that recovery-oriented staff training can impact recovery outcomes for consumers.

Nested within the PULSAR trial testing the effect of ROP staff training, the current qualitative study further examined the views of consumers receiving care in community mental health services where the REFOCUS–PULSAR staff training had been provided. The main aim of the current qualitative study was to investigate the views and understanding of ROP among consumers receiving care in these services. This paper concludes that many consumers are unaware of many recovery concepts. One way to explore if this could be improved would be through a guided conversation between staff and consumers on the concepts that promote personal recovery.

## 2. Study Design and Participants

A qualitative research design was chosen as being well suited for research that aims to develop an understanding of events or situations from the perspective of those with direct experience and knowledge of it [19]. Further, this study was conducted in a participatory manner by a research team with mental health discipline, research, consumer and carer workforce backgrounds, and a lived experience advisory group (LEAP) to guide the design, implementation, and interpretation of this study. Participatory research approaches involve the inclusion of stakeholders to address issues that affect them. The benefits of this approach result in improved decision-making, wider access to information around what does or does not work, better understanding of complex issues, and opportunities for co-learning and reflection. As a result, this approach increases transparency and offers greater authenticity to the qualitative findings [20].

All participants were adults aged 18 years or over, with sufficient English proficiency to participate in an interview, able to provide informed consent, and receiving mental health care from a publicly funded community mental health service in which staff had participated in the REFOCUS–PULSAR training. Initially, a convenience sampling approach was used in this study. Recruitment for subsequent data collection took a purposive sampling approach, informed by the current profile of consumers at the participating sites with efforts made to ensure consumers from the breadth of mental health services involved in ROP training were invited to participate.

Participants were recruited from nine participating sites across the three clinical mental health and mental health community support-service organizations involved in the PULSAR trial [17]. Initially, a convenience sampling approach was used, with information about the study being shared with consumers at services in the first round of REFOCUS–PULSAR training. Recruitment for subsequent data collection took a purposive sampling approach, informed by the current profile of consumers at the participating sites with efforts made to ensure consumers were invited to participate from the breadth of services involved in the PULSAR trial [17].

Approval was obtained from Monash University and Monash Health Human Ethics Committees prior to the research commencing, and all participants gave written informed consent to participate.

## 3. Procedures

Qualitative data were gathered using semi-structured interviews. The interview guide was developed by the qualitative research team and LEAP, with questions informed by members’ lived expertise and knowledge of literature on recovery and recovery-oriented practices (see Appendix A). Further interview prompts were informed by emerging themes from the first ten interviews conducted with participants in community mental health residential services; these prompts were included in eleven further interviews with participants recruited from non-residential community mental health settings. Individual face-to-face interviews of an average duration of 30 min were conducted by a consumer researcher in the final stages of completing their PhD with 5 years’ experience of qualitative interviews. Interviews were digitally recorded and transcribed verbatim. Any identifying information was removed and names were replaced with a pseudonym chosen by the participants, which are used throughout this paper. Transcripts were entered into NVivo 11 software to manage, sort, and analyze data. Recruitment, data collection and data analysis processes occurred iteratively.

## 4. Data Analysis

Interpretation of the interview data involved a participatory approach with the qualitative research team and LEAP, as employed in previous mental health research [21]. An iterative thematic inductive analysis was conducted using the 6 steps described by Braun and Clarke (2006) to identify semantic then latent themes within the data. Given the participatory nature of the study, once the research team had familiarized themselves with the data and generated initial codes (steps 1 and 2), patterns were then identified to capture initial thoughts of a theme (step 3) and discussed in the team. Coding was considered complete once the research team deemed there were no further themes within the data [4].

Subsequently, the themes were revised based on the research team feedback and broader input from the LEAP (step 4). At this stage, revision of the themes prompted a revision of the interview guide to refine the questions further for subsequent interviews along with emerging themes. 

Initial data analysis of the first 10 interviews identified four main themes: ‘the meaning of recovery’, ‘what services do to encourage recovery’, ‘recovery facilitators’, and ‘recovery barriers’. Following the further eleven interviews, all the interview data were examined following steps 1–4 outlined above. The additional data added depth and resulted in refinement of the themes (stage 5). Finally, quotes were incorporated to illustrate the themes.

## 5. Results

Twenty-one consumers participated in one-on-one interviews: 16 females and 5 males aged between 18–63 years (M = 44.6 years). All were receiving community mental health services where staff undertook REFOCUS–PULSAR training. Their length of mental health service use ranged from 9 months to 33 years (M = 13.33 years). Of the 21 consumers, 19 were born in Australia, 1 was born in East Timor, and another in Malaysia. Nineteen identified as non-indigenous Australians, one identified as Indigenous Australian, and one identified as Italian. Twenty spoke English as their primary language at home, and one spoke Italian. Seven participants did not complete high school. 

The four identified themes are described below: (1) recovery needs connection, (2) relationships that support recovery, (3) a better life, and (4) barriers to recovering, along with quotes from participants.

### 5.1. Theme 1. Recovery Needs Connection

Participants’ descriptions of recovery broadly aligned with a ‘normal’ life, where symptoms were managed with or without medication, or needing the right medication in order to “*function properly*” (Docki) and connect with others. Six participants described that for them recovery required “*meeting people and doing new things*” (Dimmi), “*contributing to society and the community*” (Casper), and being “*engaged*” with the community (J.A.). Connection with the community was emphasized as leading to a greater sense of purpose. For example, Janae, a residential participant, described how activities in the community assisted her: “*it gives me motivation, something to do when I wake-up*”.

Belonging to a community created a sense of meaning for participants. For most participants, broader community connections, beyond family and the mental health service, were important for their recovery. ‘Betty’ described feeling the freedom of not being judged because the group she attended did not know about her mental illness.

“… spending time with people who didn’t have a clue what I was going through was very refreshing for me, because I didn’t think that they were always judging me, I didn’t think my mums group friends were ever judging me…”

Having connections within the community was important for participants because they felt understood and supported. Participants found the service supported them to maintain their connections with the community, sometimes by assisting participants to engage in activities “*Friday morning they had a walking group*” (Fu) or by linking them to other services “*they put me in contact with a worker at [community mental health service]*” (Penny). The participants from residential services valued the service’s support to remain connected with family while they were receiving treatment. Flexibility in service delivery allowed participants to maintain family and community connections. For example, one participant described how the residential service supported him to maintain his relationship with his daughter.

“I have had my daughter come out here and that and they let her play in the arts room so they are able to adapt to my needs, so that is good”.(David)

Another participant described the level of support and flexibility demonstrated by her community worker ensured she maintained connections over Christmas and New Year: “*everybody goes away and you feel abandoned … she [the community worker] worked it out so that a group of us weren’t abandoned and we were able to enjoy each other’s company for a movie day*” (Sally).

Participants from residential services talked about the importance of feeling connected to the service as a whole, rather than connections and relationships with individual staff members. They described the service as a “*home-kind of environment*” (Janae), “*warm and welcoming*” (Melinda), and the service instilling a sense of freedom, including freedom to return to the service whenever they became unwell.

Many participants thought they were alone in their mental health issues. On initial admission to the service, participants were surprised that there were other people in similar situations to them. They described the unique opportunity to connect to, learn from, and help peers because of a deep understanding of common issues. “*A lot of us help each other because we are like ‘oh I had that problem and I did it this way’ and they go ‘oh yeah ok let’s do that’. So we all help each other so that is the best thing about [the residential service] …*” (Villanous). For a small number of participants, having connection or friends with mental health issues was something to “*steer clear*” of with a preference to connect with one’s own friends, family, and community. ‘Velvet’ described her perceptions: “*I steer clear from people that do have mental illness, only because they’re usually a lot sicker than I am and have a range of problems and issues and they’re harder to deal with*”.

### 5.2. Theme 2. Relationships That Support Recovery

Similarly, to residential participants, others in the community described the importance of connection and their relationships with staff members. Participants reported the value of staff that understood them and who they could talk to. Janae described “*talking it out [problems] with them… what is going on for me… I feel more relief [from her anxiety]*”. Three participants described situations where they believed that staff acted beyond their job requirements to demonstrate friendship and understanding, as described by ‘Dimmi’:

“Well, they’re really caring and I don’t know whether this matters, but I happened to mention that I really like bath products, and when my worker came around, she brought a big basket of bath products that the [service] staff had put together for me”.

Participants valued the support from staff, and they looked to them for guidance, help with coping strategies, and problem solving. Participants appreciated the willingness of staff to listen and allow them to vent their feelings, whilst providing support and coping strategies. For example, participants at the residential services reported that staff eagerness and enthusiasm made a difference to participants’ engagement in activities and goals. 

“The eagerness that they do it with. It’s not like ‘oh you want to go bush walking’ they really get in to it and you can have a deep and meaningful conversation”.(Docki)

Participants reported that positive relationships with staff helped them to remain engaged with the service and supported their recovery. Janae described how “*she [her worker] made me feel like she was my best friend sitting across the table… she made me feel comfortable that I felt free talking to her, and now she’s opened me up to being myself”.*

Through these relationships, meaningful conversations occurred in particular for those in residential setting. For example, one participant described discussions about recovery in group sessions was focused more around education, such as coping strategies, drug and alcohol use, and meditation. Subsequently, three participants from these services talked about methods to support recovery… 

“I have only had one one-on-one time and it was with the recovery book. And I just needed to figure out like green light is when I am well and orange light is when I am getting unwell and red [light] is when I am very unwell….and I need to call someone”.(Janae)

Seven participants were unable to identify direct discussion with staff about recovery. However, of these, three participants reported that staff had talked about recovery concepts or similar ideas, but they had not directly used the word ‘recovery’. Casper described how staff “*got me to do some goal setting*” and Docki who *described “they basically talked to you about your interests … I would never have done that before … I came here they actually talked to me and helped me–I realized how much I liked it … it makes you think well, like life can be better than it was before I came in [to the residential setting]*”.

From their perspective, participants commented that staff could sympathize, but they did not always understand what consumers were going through.

“[Staff] don’t understand what I am going through, don’t understand what other people are going through and can’t sympathise and can’t…um they can sympathise in the way they see people going through such distressing things but they can’t help you through it. They can’t say well I went through this and I did this to help”.(Villanous)

Although some participants felt being with around peers (in the residential setting) supported a unique connection “*you have this stigma about mental health in here [residential setting] so we are all trying to help each other*” (Janae), not all participants viewed being in the presence of peers as positive or took up the opportunity. ‘Betty’ declined because she thought they might make her feel worse. 

“They offered me to go into not necessarily a group meeting, but like a little community meeting where other people going through what was I was going through were there, and I decided against that because I thought that might make me crazy”.

Hence, suggesting that the sources stigma and discrimination faced can be identified among consumers as well as staff.

### 5.3. Theme 3. A Better Life

For many, building a satisfying life was their view of recovery. Participants typically described these aspects as representing an “*ideal lifestyle*” that included “*a good relationship*”, “*work*” or “study”, and “*hobbies*”. Other participants identified recovery as entailing personal growth, typically in regard to building their confidence and identity. These participants described it as a time to “*get over that grieving pain*” (Penny), or enjoying a time when they did not feel bewildered in their experience of mental illness. ‘Joey B’ emphasized recovery as a long-term effort of stepping forward, and building a better self. 

“Recovery to me is like getting better…. Or just like stepping forward from where you used to be and building a better self…. better than your old self… it’s not easy, and it is not something that can be quickly done. It takes a lot of time”.

Additionally, participants acknowledged that the service’s staff were approachable and willing to talk, but that regular supportive one-on-one sessions were also needed. 

“… more one on one so if they go … ‘hey come in this room and we are going to help you talk about your anxiety and coping strategies’… I would like more of that... more of that one-on-one”.(Janae)

Participants described the relentless weight and struggle from living with mental health issues. However, they had hope in the service to help them to recover. Hope was inspired by the participants’ perception of available support, such as knowing they could speak to staff whenever they needed. Hope was also described by a participant when she stated that the staff “*always knew how to keep* [her] *in the forward motion*” (Betty). For one participant, hope was simply by way of sharing the day:

“So, this is definitely the way… [the service] works is that there is hope out there and hope in here in that you are sharing your day and doing your things”.(Melinda)

The residential service supported consumers to create structure in their daily life, which they found helpful. Participants wanted to know what was happening each day so they could use their time effectively, although, at times, this information was lacking. Participants also appreciated the help of staff in planning their time away from the service. The service supported participants to maintain their existing commitments, whilst opening them up to new things. As one residential participant described …

“…it is a really positive environment and you are socializing with people and you learning the basic necessities of life because when I get home I kind of fall out of the habit of that whereas like here it is your responsibility to cook. You are up at a decent hour of the day… the programs really allow you to explore different hobbies and opens you up to new things”.(Bailey)

### 5.4. Theme 4. Barriers to Recovering

Not all aspects of service use were conducive to recovery. Participants identified two main barriers to recovery: the first being the nature of their mental health issue such as being medication focused and having limited treatment options; the second being challenges with the mental health service, such as disrupted or difficult relationships with staff, changes in staff allocation, or not being able to meet individual needs. However, both of these barriers had some degree of overlap based on participants’ views.

Many participants thought that their services had an over-reliance on medication as the primary treatment option for a “*bad day*” rather than other meaningful activity as can be seen in the following quote from a residential participant:

“There is nothing to get someone though a bad day–they just have to get [you] through the bad day and live it out… it is just up the medication and give me some Valium you know…so there is no practices like that…I don’t know what I would suggest…”.(Janae)

Participants wanted other treatment options that included assistance and time to develop their coping skills and strategies for managing symptoms. In the residential service, participants thought a staff member should be onsite always to help problem solve issues and in the case of emergency.

This idea was supported by other participants who reported that although staff supported them to have opinions about their treatment, ultimately there were limited options. This was frequently attributed to the doctor or psychiatrist not taking the time to explore other options. Limited opportunity to talk to the doctor meant that the interactions were mostly about medication management.

“I would like a longer period [of time] to see your psychiatric doctor when you are in here …They only ask questions like how are you sleeping and what is your medication… and in general how are you doing. And so they basically write out your script or something like that whereas you might to have some more one-on-one time …”.(Melinda)

The second main barrier was in relation the service provision as a whole and/or the staff within the service. Just as a good relationship with staff assisted participants to connect to the service and support their recovery, a disrupted or difficult relationship had a negative impact on consumer engagement. Nine participants described staff who they felt either did not understand their needs, provide enough support, or were regularly unavailable.

“She [the worker] was always on holidays. I got to see her about 3 times…I don’t think I saw her in the last 6 months. She’s always overseas”.(Penny)

Moreover, participants reported not having a say in the allocation of staff, including having a choice in gender. For ‘Betty’, this was not a problem, but she envisaged that it might be for some people.

“… there is one thing that I noticed, that it wasn’t necessarily a bad thing, but it could have been I guess in some cases, but no women came, it was all men, and I don’t know if that’s just because there are only men on that particular team that came to me… I understand for safety concerns that they may not have sent women… So always two men… which I didn’t have a problem with, but it was something I noticed”. 

A similar barrier was evident in residential settings whereby the residential service did not support the needs of all the participants. For example, one participant explained that he was not interested in group activities, and described the impact of a lack of meaningful things to do when staying at the service. 

Finally, a minor, but important overarching theme was evident across all the data. This was expressed as an ‘uncertainty about the future’. This uncertainty was the cause of concern for a number of participants. Overwhelmingly, participants’ descriptions of their experiences of mental health issues reflected a tone of a being “*weighed down*” by a relentless struggle to achieve recovery. One participant described being unwell with a mental health issue as being a “*failure*”. Participants were unsure of what their futures held and, at times, were uncertain as to the direction of their treatment. For instance, one participant described how she had been unwell for so long that she could not remember what “*normal*” was. Subsequently, she was unsure of the outcome for which she was aiming.

“At the end of my stay here what am I meant to be? Will I be well or will I be unwell and end up back here? That is my fear. … They say if you can’t cope at home now what is going to happen in 4 weeks when you go home. And I don’t know what is going to happen”(Janae)

## 6. Discussion

The current qualitative study investigated the views of consumers in a community mental health service as part the REFOCUS–PULSAR trial, which showed that recovery-oriented training of staff can lead to improvement in consumers’ personal recovery. Overall, the findings for the current study highlight the need for staff training to support a collaborative and recovery-oriented approach to care, which aligns with prior research [16].

Across all consumers there were four main themes. These were: ‘recovery needs connection’, ‘relationships that support recovery’, ‘a better life’, and ‘barriers to recovering’. In addition, a minor, but overarching theme arose from the data, which was ‘uncertainty about the future.’ Each theme was strongly interlinked.

The concept of ‘recovery needed connection’ aligns with prior research whereby connection can come in many forms to support recovery such as through relationships [5,15]. In this study, the concept of connection was described as a broader idea for participants rather than being based on individual people. Those in residential settings focused on forming connections within the group setting whilst looking ahead to being a part of the broader community and building connections in the future. For those receiving treatment in the community, a connection to others within the community was highly valued as supporting their recovery.

A related theme was ‘relationships that support recovery’; however, for this theme, participants focused on an individual relationship that gave meaning to their recovery. Staff were considered as important to manage stress, listen, and support routines to create a sense of purpose. In contrast, although some staff were seen as supportive, others found staff turnover to be problematic resulting in a lack of continuity, lack of access in times of need, and at times staff being slow to act. This finding supports the prior work of Lau and Hutchinson (2021), and Wood and Alsawy (2015), who found that relationships with mental health staff can either support or hinder recovery. This can be a particular challenge if consumers do not develop a trusting, collaborative relationship with professionals [10,11,16].

For many, within the current study, the focus was on building a better, more satisfying life. What constituted a better ‘life’ was varied for the participants highlighting that the concept of recovery as being highly individual [4]. For many consumers, the process of recovery was how they made meaning of their experience, which included aspects such as developing a sense of purpose, fostering hope and a new future, or developing relationships that can support their wellbeing through the recovery journey [5,8,10]. The tools introduced in ROP training for staff focuses on values, strengths, and goal striving, which are considered as important aspects in the recovery journey. In a prior study exploring consumers’ experience of service use, after similar staff ROP training in the UK, consumers reported that collaborating with staff on those areas was empowering and hope-inspiring [16]. As such the importance of staff and their approach to supporting personal recovery has implications in managing the care of those with mental health issues.

The final theme was participant perceptions of barriers to recovery with sub-themes on the nature of having a diagnosis of a mental health issue and the service provision to support them in their recovery journey. The findings highlight the role that mental health services have in supporting consumers’ personal recovery journey. Services have an important role in creating spaces and processes that support relationships between consumer and staff in order to grow hope and meaning. Without processes and structures that support a recovery relationship, services risk creating barriers, disrupting relationships, and increasing uncertainty in the future. When consumers feel welcome and supported in a service, it can often be attributed to the strength of a healthy relationship with professionals in that service [16].

Despite each of these themes, for some participants there was an overarching theme around a strong sense of uncertainty for the future. This expression of uncertainty, doubt, and “*fear*” suggested that participants’ sense of recovery and what their future would look like was unclear to them. Given the personal nature of recovery, it is possible that these individuals have yet to find meaning in the process or that their experience within the mental health setting has yet to bridge the gap between staff and consumer views of recovery [3,5,15,16].

Throughout this study, it was evident that only a minority of consumers could identify particular ROP language, for example, working on a strength; the vast majority being unable to recall any aspects of recovery despite being prompted about their personal experiences. This finding aligns with the study by Wallace et al. (2016) who similarly found some individuals were unable to recall any aspects of the intervention. This lack of recall being attributed to the ‘subtle changes in the working relationship’ (p. 1282). Similar to the study by Wallace et al. (2016), it is possible in the current study that participants were unable to differentiate any change or difference perhaps due to the duration of time with the service, especially given some participants were in a short-term residential setting. However, in the current study the second iteration of interviews included questions directed to illicit participants’ knowledge with prompting questions and a reflection on prior themes. Despite this inclusion, there was still limited understanding of ROP terminology. This suggests that services need to enhance consumers’ awareness on ROP when it is implemented and being used via direct facilitated conversation perhaps with the use of a recovery handbook. This open-facilitate approach is at the heart of ROP care, which aims to enhance collaborative care through co-ownership [16].

Our findings highlight how the mechanism of ROP training, when successful, supports the consumer’s journey to view their lives in a more holistic manner. The components of this training included connection to the community, self-care, and factors such as work and housing. Consumers in the current study also recognized the importance of medication or the right medications to assist them in the recovery. These features align with a scoping review by Gyamfi and colleagues (2022) who suggest that the recovery consists of both internal processes such as coping and hope, and external factors such as social supports and appropriate housing. Although these aspects may be considered as fundamental components to ROP, understanding the underlying mechanism that supports recovery appears to be a key factor to ROP success [8].

This study had several strengths and limitations. The main strength lies in consumer involvement in research design, data collection, and analysis to examine the under-researched area of consumer views and experiences of ROP training of staff in mental health services. In addition, a further strength was the participatory nature of the study, which aimed to conduct two sets of interviews: the first being to gather initial data; and the second to reflect on deidentified themes in order to be able to illicit greater depth in participant responses around ROP, as well as how improvement could be made to support staff when they are supporting consumers in their recovery. One of the main limitations was that consumers were asked to recall experiences that occurred potentially in times of distress and difficulty. One means to overcome this may be through the use of focus groups to support recall and the sharing of experiences and views. Future research could also consider recruiting consumer/staff dyads to explore aspects of their relationship in working towards recovery, as well as examining consumers’ experiences of recovery using a longitudinal approach.

## 7. Conclusions

This qualitative study aimed to shed light on consumer views of recovery following staff training in ROP. While the importance of consumer/staff relationships was a prominent finding of this study, the meaning that consumers attribute to their recovery was perhaps more important. A strong relationship between staff and consumers provided hope for recovery, and assisted them to build meaning in their life through exploring possibilities. However, consumers placed great emphasis on the need to explore aspects of their lives that they felt enhanced their recovery, such as the right medication, connection to the community, a sense of purpose, and building a satisfying life (whatever that meant to them). Some negative consequences were noted by consumers when relationships with staff were disrupted or not a good fit, such as disengagement from the service, but there were also positive consequences such as strengthening connections.

This study adds to the growing body of knowledge on the training of mental health care staff in ROP and the consumer experience following that training. However, given the apparent lack of knowledge shown by consumers on the concepts of ROP, it could be suggested that clearer awareness using a facilitated conversation may be of benefit. It is recommended that a facilitated conversation should include the use of a specifically designed ‘recovery’ handbook to aid in this process. It is evident that an awareness of recovery using ROP for consumers is dependent upon many varied and individual factors. In order to support consumers in their recovery journey, there is a need to ensure mental health care staff use a consistent, collaborative approach with more overt conversations on recovery to enhance the consumer experience.

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
