# Peer review of "Consumer Views and Experiences of Secondary-Care Services Following REFOCUS-PULSAR Staff Recovery-Oriented Practices Training"

_ijerph, 2023, doi:10.3390/ijerph20105894_

Round 1

Reviewer 1 Report

1.     The article is interesting.

2.     The research is valuable due to the topic, which is an important issue.

3.     This is a meaningful study as this manuscript contributes to qualitative analysis.

4.     From the perspective of this reader, one suggestion is to add some objective statements from the result.

Author Response

Reviewer comment

Author response

Reviewer 1

 1. The article is interesting.

2.     The research is valuable due to the topic, which is an important issue.

3.     This is a meaningful study as this manuscript contributes to qualitative analysis.

4.     From the perspective of this reader, one suggestion is to add some objective statements from the result.

Thank you for your feedback. In response to point 4, we appreciate this suggestion. We are unclear what additional statements would be helpful as there are several statements about the results included at lines 193-195, 215-218, 280-289 of the manuscript.

Reviewer 2 Report

The aim of the author’s study was to understand the value and impact of ROP training from a service users perspective. It is relevant and addresses a specific gap - service users perspective.

Methods: Controls are not relevant - the methods used as congruent with the chosen methodology.

The conclusions are consistent and they address the main question posed, in addition they propose a way forward  recommendations, the references are appropiate.

Fgures and tables:  N/A, however the use of quotes/themes are approprate.

An interesting study, which gives a robust service user insight into ROP and then impact of training. The recommendations signpost a way forward for this type of training off, one that is user-centric.

Author Response

Reviewer Comment

Response

Reviewer 2

1.       The aim of the author’s study was to understand the value and impact of ROP training from a service users perspective. It is relevant and addresses a specific gap - service users perspective.

2.       Methods: Controls are not relevant - the methods used as congruent with the chosen methodology.

3.       The conclusions are consistent and they address the main question posed, in addition they propose a way forward  recommendations, the references are appropriate.

4.       Figures and tables:  N/A, however the use of quotes/themes are appropriate.

5.       An interesting study, which gives a robust service user insight into ROP and then impact of training. The recommendations signpost a way forward for this type of training off, one that is user-centric.

Thank you for your comments

Reviewer 3 Report

The study qualitatively investigated consumers' perspectives on community mental health services where REFOCUS-PULSAR training was provided. Overall, the manuscript is well-written, with a few minor points for review.

1.      Study Aim

    The study's aim is somewhat unclear. Did the authors intend to simply explore consumers' views on community mental health services, or did they seek to investigate how consumers perceived "recovery" and their experiences when receiving recovery-oriented services? I recommended that the authors clarify their aim at the end of the Background section.

2.      Sampling

   I suggest that the authors provide more detailed information on the sampling procedure. How many teams or areas were involved in participant recruitment? Additionally, it would be helpful to include the number of people who declined to participate in this research.

3.      Interviews and Expertise

    In this study, a consumer researcher conducted face-to-face interviews. I suggest that the authors clarify whether this researcher had prior experience conducting qualitative interviews and received training in qualitative research.

Author Response

Reviewer comment

Response

The study qualitatively investigated consumers' perspectives on community mental health services where REFOCUS-PULSAR training was provided. Overall, the manuscript is well-written, with a few minor points for review.

 1.      Study Aim

    The study's aim is somewhat unclear. Did the authors intend to simply explore consumers' views on community mental health services, or did they seek to investigate how consumers perceived "recovery" and their experiences when receiving recovery-oriented services? I recommended that the authors clarify their aim at the end of the Background section.

 2.      Sampling

   I suggest that the authors provide more detailed information on the sampling procedure. How many teams or areas were involved in participant recruitment? Additionally, it would be helpful to include the number of people who declined to participate in this research.

 3.      Interviews and Expertise

    In this study, a consumer researcher conducted face-to-face interviews. I suggest that the authors clarify whether this researcher had prior experience conducting qualitative interviews and received training in qualitative research.

1. Amendments have been made for clarity and wording changes commencing at line 101.

Nested within the PULSAR trial testing the effect of ROP staff training, the current qualitative study further examined the views of consumers receiving care in community mental health services where the REFOCUS-PULSAR staff training had been provided. The main aim of the current qualitative study was to investigate the views and understanding of ROP among consumers receiving care in these services. This paper concludes that many consumers are unaware of many recovery concepts. One way to explore if this could be improved would be through a guided conversation between staff and consumers on the concepts that promote personal recovery.

2. Sampling - Additional information on the recruitment has been added at line 135

Participants were recruited from nine participating sites across the three clinical mental health and mental health community support service organisations involved in the PULSAR trial (Meadows et al., 2019). Initially, a convenience sampling approach was used, with information about the study being shared with consumers at services in the first round of REFOCUS-PULSAR training. Recruitment for subsequent data collection took a purposive sampling approach, informed by the current profile of consumers at the participating sites with efforts made to ensure consumers were invited to participate from the breadth of services involved in the PULSAR trial (Meadows et al., 2019). 

3. Interviewer expertise -  The following has been added at line 154

‘were completed by a consumer researcher in the final stages of completing their PhD, with 5 years' experience of qualitative interviewing’.